# Development of a Monoclonal Antibody Targeting HTLV-1 Envelope gp46 Glycoprotein and Its Application to Near-Infrared Photoimmuno-Antimicrobial Strategy

**DOI:** 10.3390/v14102153

**Published:** 2022-09-29

**Authors:** Yasuyoshi Hatayama, Yutaro Yamaoka, Takeshi Morita, Sundararaj Stanleyraj Jeremiah, Kei Miyakawa, Mayuko Nishi, Yayoi Kimura, Makoto Mitsunaga, Tadayuki Iwase, Hirokazu Kimura, Naoki Yamamoto, Akifumi Takaori-Kondo, Hideki Hasegawa, Akihide Ryo

**Affiliations:** 1Department of Microbiology, Yokohama City University School of Medicine, Yokohama 236-0004, Japan; 2Life Science Laboratory, Technology and Development Division, Kanto Chemical Co., Inc., Isehara 259-1146, Japan; 3Advanced Medical Research Center, Yokohama City University, Yokohama 236-0004, Japan; 4Division of Gastroenterology and Hepatology, Department of Internal Medicine, The Jikei University School of Medicine, Minato 105-8461, Japan; 5Research Center for Medical Sciences, The Jikei University School of Medicine, 3-25-8, Minato 105-8461, Japan; 6Department of Health Science, Gunma Paz University Graduate School, Takasaki 370-0006, Japan; 7Genome Medical Sciences Project, National Center for Global Health and Medicine, Ichikawa 272-8516, Japan; 8Department of Hematology and Oncology, Graduate School of Medicine, Kyoto University, Kyoto 606-8501, Japan; 9Center for Influenza and Respiratory Virus Research, National Institute of Infectious Diseases, Musashimurayama 208-0011, Japan

**Keywords:** HTLV-1, monoclonal antibody, near infrared photoimmuno-antimicrobial strategy, IR700

## Abstract

Human T-cell leukemia virus type 1 (HTLV-1), a retrovirus, causes adult T-cell leukemia-lymphoma, HTLV-1 associated myelopathy/tropical spastic paraparesis, and HTLV-1 uveitis. Currently, no antiretroviral therapies or vaccines are available for HTLV-1 infection. This study aimed to develop an antibody against the HTLV-1 envelope protein (Env) and apply it to a near-infrared photoimmuno-antimicrobial strategy (NIR-PIAS) to eliminate HTLV-1 infected cells. We established mouse monoclonal antibodies (mAbs) against HTLV-1 Env by immunization with a complex of liposome and the recombinant protein. Detailed epitope mapping revealed that one of the mAbs bound to the proline-rich region of gp46 and exhibited no obvious neutralizing activity to inhibit viral infection. Instead, the mAb was rarely internalized intracellularly and remained on the cell surface of HTLV-1-infected cells. The antibody conjugated to the photosensitive dye IRDye700Dx recognized HTLV-1 infected cells and killed them following NIR irradiation. These results suggest that the novel mAb and NIR-PIAS could be developed as a new targeted therapeutic tool against HTLV-1 infected cells.

## 1. Introduction

Human T-cell leukemia virus type 1 (HTLV-1) was the first retrovirus discovered in humans and is known to cause adult T-cell leukemia-lymphoma (ATL) and other HTLV-1-related diseases, including HTLV-1-associated myelopathy/tropical spastic paraparesis (HAM/TSP), and HTLV-1 uveitis (HU) [1,2,3]. It is estimated that 5–10 million people worldwide are chronically infected with HTLV-1 [4], with ATL occurring in 5% [5], HU in 0.1–0.8% [6,7], and HAM/TSP in 0.25–4% [8] of patients after several decades of latent infection. ATL is classified into four subtypes according to clinical symptoms, i.e., acute, lymphoma, chronic, and smoldering [9]. The overall 4-year survival rate for each subtype is 11%, 16%, 36%, and 52%, respectively, and ATL is still difficult to treat [10].

HTLV-1 is a positive-sense, enveloped, single-stranded RNA virus that mainly infects CD4-positive T-cells and causes latent infection by integrating into the host genome via virus integrase [11]. The major replication pathway of HTLV-1 is cell-to-cell infection, which involves two glycoproteins on the envelope (Env). Cell surface protein gp46 and transmembrane protein gp21 are synthesized from the cleaved precursor protein gp62, and they function as a trimer to facilitate viral entry [12,13]. Once HTLV-1 enters the target cell via cell fusion, the RNA genome is reverse transcribed, and its DNA product is inserted into the host genome. The provirus is modified after transcription by RNA polymerase II in the cell. The viral proteins Gag, Pol, and Env are then translated. These viral proteins and genomic RNA assemble to form immature viral particles, which are matured by viral proteases to form infectious viral particles [14,15].

HTLV-1 is transmitted via body fluids, including breast milk, semen, and blood, spreading via cell-to-cell infection. Vertical mother-to-child transmission (MTCT) through breastfeeding is the predominant route of HTLV-1 infection [16,17,18]. Horizontal transmission of HTLV-1 occurs through sexual intercourse. Acquiring HTLV-1 early in life is thought to be a potential risk factor for developing ATL [17]. Current methods to prevent MTCT include weaning, short-term breastfeeding for 3 months, and the use of processed milk. However, these methods are not sufficient in preventing MTCT. Therefore, reliable treatments for HTLV-1 infection need to be developed.

Mutations in the Env protein of HTLV-1 rarely occur, unlike in human immunodeficiency virus (HIV) [19]. Therefore, it is possible to use monoclonal antibodies (mAb) against the HTLV-1 Env to prevent the infection and progression of diseases caused by HTLV-1. The anti-gp46 mAb is capable of blocking viral infection by the direct neutralization and eradication of HTLV-1 infected cells via antibody-dependent cellular cytotoxicity (ADCC) [20,21]. Although anti-gp46 mAb can block horizontal infection in humanized mice [22], there is no effective antibody drug that can target and eliminate HTLV-1 infected cells.

Recently, near-infrared photoimmunotherapy (NIR-PIT) has been developed as a new treatment for cancer. It specifically destroys cancer cells without damaging normal cells and tissues. This technology uses target-specific mAbs conjugated to the phthalocyanine dye IRDye700Dx (IR700), which is photosensitive to NIR [23]. When IR700 is exposed to NIR light, axial ligands attached to the silicon atom at the core of phthalocyanine dissociate with hydrolysis. The axial ligand-release reaction of IR700 leads to physical changes within the mAb-antigen complex, which in turn causes cell membrane damage to target cells. Accumulation of such events on the cellular membrane induces weakness in membrane integrity, leading to the bursting associated with photoimmunotherapy and resulting in cell death [24]. In addition, NIR-PIT also possesses indirect anticancer effects through the activation of host immunity by releasing multiple cellular compounds and antigens from the destroyed cancer cells. The anticancer action of NIR-PIT has been demonstrated in many pre-clinical experiments in gastric cancer [25,26], primary lung cancer [27], small cell cancer [28], breast cancer [29,30], hepatocellular carcinoma [31], glioblastoma [32,33], melanoma [34], and B cell lymphoma [35,36]. In addition to cancer treatment, it has also been applied to pathogenic microorganisms. We recently demonstrated a photoimmuno-antimicrobial strategy (PIAS) that eliminated antibody targets using a photo-activated anti-pathogen antibody. Indeed, PIAS can target several microorganisms such as the fungus *Candida albicans* [37,38], the bacterium-methicillin-resistant *Staphylococcus aureus* (MRSA) [37], the bacteriophage [39], and HIV-1 Env-expressing cells [40]. In this study, we developed an mAb against HTLV-1 Env and investigated the application of PIAS for antibody-based killing of HTLV-1-infected cells.

## 2. Materials and Methods

### 2.1. Cells

Jurkat (clone E6.1, ATCC TIB-152), MT-2 cells (JCRB 1210), and MT-1 cells (JCRB 1209) were maintained in RPMI-1640 medium (FUJIFILM Wako Pure Chemical Corporation, Osaka, Japan) containing 10% (*w*/*v*) heat-inactivated fetal bovine serum (FBS) (Gibco, Waltham, MA, USA). HTLV-1 long terminal repeat (LTR) driven secreted NanoLuc luciferase (LTR-secNL) Jurkat cells were selected and maintained in RPMI-1640 medium supplemented with 10% (*w*/*v*) heat-inactivated FBS and 1 μg/mL puromycin. Anti-HTLV-1 Env antibody-generating hybridomas were cultured in GIT medium (FUJIFILM Wako Pure Chemical Corporation) with human-recombinant IL-6 (R&D SYSTEMS, Minneapolis, MN, USA). All cells were incubated in 5% CO_2_ at 37 °C.

### 2.2. Plasmid Construction

A codon-optimized artificial gene was synthesized from the amino acid sequence of HTLV-1 Env (strain Japan ATK-1 subtype A; UniprotKB_P03381) by Eurofins Scientific SE (Val Fleuri, Luxembourg). The synthetic gene was cloned into pEU-E01- Histidine (His)-TEV-MCS, pEU-E01-His-TEV-MCS-bls (CellFree Sciences, Yokohama, Japan) using Xho1 and Not1. Deletion mutants of HTLV-1 Env for epitope analysis were engineered using the PrimeSTAR Mutagenesis Basal Kit (Takara Bio Inc., Kusatsu, Japan). The HTLV-1 LTR (GenBank_Z31660.1) was also codon-optimized and synthesized by Eurofins Genomics. The synthetic gene was restriction cloned into pGF1-MCs-secNL using EcoR1 and Xho1. The pGF1-MCs-secNL backbone was generated by inserting the secreted NanoLuc reporter gene (Promega Corporation, Madison, WI, USA) into the pGF1 vector (System Biosciences, Palo Alto, CA, USA).

### 2.3. Cell-Free Protein Synthesis and Purification

In vitro wheat germ cell-free protein synthesis was performed as previously described [41,42]. For mouse immunization, in vitro transcription was performed using SP6 polymerase. A dialysis cup was filled with a reaction mixture containing WEPRO7240 (CellFree Sciences) wheat germ extract, 250 µL mRNA, 40 µg/mL creatinine kinase, and 10 mg/mL azolectin liposomes. The mixture was overlaid with 5.5 mL of the SUB-AMIX SGC solution (CellFree Sciences) and then incubated at 15 °C for 72 h using the dialysis overlay method. Here, the upper dialysis cup was placed in a vessel containing 40 mL of SUB-AMIX SGC solution. The synthesized His-Tev-HTLV-1 Env protein was purified using the Accudenz concentration gradient method [43]. After isolating the HTLV-1 Env protein anchored in liposomes, the buffer was replaced with phosphate-buffered saline (PBS). Subsequently, purified proteins were confirmed by Coomassie brilliant blue (CBB) staining using RapidCBB KANTO 3S (Kanto Chemical, Tokyo, Japan). The other proteins were synthesized using the bilayer translation method without liposomes [41,42]. The synthesized proteins were purified using Nickel-Sepharose high-performance beads (GE Healthcare, Waukesha, WI, USA). Thereafter, the bound proteins were eluted with elution buffer [20 mM Tris-HCl (pH 7.5), 500 mM NaCl, 500 mM imidazole], and the purified protein was confirmed by immunoblotting using horseradish peroxidase (HRP)-conjugated streptavidin (1:1000 dilution, #RPN1231v; GE Healthcare).

### 2.4. Immunization and Generation of Hybridomas

Immunization of BALB/c mice and the generation of hybridomas were carried out as previously described [44,45]. Briefly, the BALB/c mice were immunized by injecting purified HTLV-1 Env protein. Four weeks after immunization, the splenocytes of the mice were fused with myeloma cells, and four hybridoma cells were established.

### 2.5. Purification of mAbs

Hybridoma cells were grown in CD Hybridoma and AGT mediums (Thermo Fisher Scientific, Waltham, MA, USA). Antibodies in the culture medium of each clone were separated by centrifugation at 8000 rpm for 15 min and purified on AcroSep Hyper DF columns (Pall Corporation, Port Washington, NY, USA). The concentration of the purified mAbs was calculated by measuring the absorbance at OD_280_. Immunoglobulin characterization was performed using an IsoStrip mouse monoclonal antibody isotyping kit (Roche Diagnostics, Mannheim, Germany).

### 2.6. Flow Cytometry Analysis

When the MT-2 and Jurkat cell cultures reached a density of 1 × 10^6^ cells/mL, 10 μg/mL mAbs (clone A–D) were added for 16 h at 4 °C. Next, the cells were washed twice with wash buffer [containing 2% (*v*/*v*) FBS and 0.02% (*w*/*v*) NaN_3_ in PBS] and incubated with phycoerythrin (PE) goat anti-mouse IgG (BioLegend, San Diego, CA, USA) as a secondary antibody for 1 h on ice under light. Next, the cells were washed twice with wash buffer. Finally, the cells were fixed in 4% (*v*/*v*) paraformaldehyde phosphate buffer (FUJIFILM Wako Pure Chemical Corporation) and subjected to fluorescence-activated cell sorting (FACS) analysis using a BD FACS Canto TM II Flow Cytometer (BD Biosciences, Franklin Lakes, NJ, USA).

### 2.7. Immunofluorescence (IF)

For immunofluorescence, 5 × 10^5^ cells/mL of MT-2 and Jurkat cells were washed with PBS, fixed in 4% (*v*/*v*) paraformaldehyde phosphate buffer, and washed again with PBS. After blocking with 2% (*w*/*v*) bovine serum albumin (BSA) in PBS for 1 h at room temperature (RT), the cells were washed three times with PBS and incubated with primary antibody diluted in 0.1% (*w*/*v*) BSA in PBS overnight at 4 °C. The cells were washed three times with PBS and incubated with goat anti-mouse IgG, human adε-FITC (Beckman Coulter, Brea, CA, USA) diluted in 0.1% (*w*/*v*) BSA in PBS for 1 h at RT. The cells were washed three times with PBS and incubated with Prolong Gold antifade reagent containing DAPI (Invitrogen, Carlsbad, CA, USA). Microscopic observations were performed using a BZ-9000 fluorescence microscope (Keyence, Osaka, Japan).

### 2.8. Enzyme-Linked Immunosorbent Assay (ELISA)

An HTLV-1 Env recombinant protein solution was diluted in 50 mM carbonate buffer (pH 9.6) and added to ELISA plates (Thermo Fisher Scientific). To immobilize the proteins, the plates were incubated overnight at 4 °C. Subsequently, the wells were blocked with 2% (*w*/*v*) skim milk in PBS containing 0.05% (*v*/*v*) Tween-20 (PBS-T) for 1 h at RT and washed three times with PBS-T. Next, 50 μL of purified antibody diluted to the desired concentration in PBS-T was added to each well, and the plates were incubated for 1 h at 37 °C. After washing three times with PBS-T, 100 μL of anti-mouse IgG conjugated to HRP were added to each well, and the plates were incubated at 37 °C for 1 h. After washing three times with PBS-T, 100 μL of the TMB substrate (Kirkegaard & Perry Laboratories, Washington, DC, USA) was added and absorbance was measured at 450 nm using the GloMax Discover System (Promega).

### 2.9. AlphaScreen Analysis

Here, 1.5 μL of 1 μM N-terminal biotinylated peptides (PRD1-PRD10) and 0.1 μL mAb were mixed in a 384-well AlphaScreen plate (PerkinElmer, Boston, MA, USA) in AlphaScreen Buffer containing 20 mM Tris-HCl (pH 7.6), 0.01% (*v*/*v*) Tween-20, and 1 mg/mL BSA. The binding reaction was carried out at 26 °C for 30 min. Next, protein G conjugated acceptor beads and streptavidin-coated donor beads were added to the solution, and the binding reaction was carried out at 26 °C for 1 h. AlphaScreen signals were measured using an EnVision plate reader (PerkinElmer). The anti-His mAb was used as an mAb negative control, and biotinylated DHFR (dihydrofolate reductase) was used as a peptide negative control.

### 2.10. Epitope Prediction

The three-dimensional structure prediction model of the HTLV-1 gp46 protein was created by inputting the amino acid sequence of gp46 (1–312 amino acid of HTLV-1 Env) into AlphaFold 2 [46]. The binding position of the mAb Clone D epitope was highlighted by UCSF Chimera [47].

### 2.11. Affinity Measurement of mAbs

Bio-Layer Interferometry (BLI) was used to measure the equilibrium dissociation constant (K_D_) values using an Octet RED96 system (ForteBio, Fremont, CA, USA), as previously reported [48]. MAb Clone D (20 µg/mL) was loaded onto an anti-mouse IgG capture biosensor chip (ForteBio) containing PBS supplemented with 0.1% (*w*/*v*) BSA and 0.01% (*v*/*v*) Tween-20 for 5 min. The association of recombinant gp46 at concentrations of 200, 100, 50, 25, and 12.5 nM was measured for 5 min, followed by a 10-min-long dissociation phase. All measurements were corrected for baseline drift by subtracting the reference values. The operating temperature was maintained at 30 °C. Data were analyzed using a 1:1 coupled model with a global fitting algorithm using the ForteBio data analysis software.

### 2.12. Reporter Cells Generation and Reporter Assay 

A new system for assessing viral transmission was developed by modifying the method described by Alais et al. [49]. To generate Jurkat LTR-secNL cells, LentiX-293T cells seeded in a 6 cm dish were transfected with pGF1-LTR-secNL, HIV Gag-pol, and VSV-G, using Lipofectamine 3000. Lentiviruses were generated by collecting the supernatants after 48 h. The lentiviruses were mixed with Jurkat E6.1 cells and incubated for 24 h for transfection. Next, puromycin (1 ng/mL) was added to select the infected cells.

LAT-27, a neutralizing antibody for HTLV-1, was provided by Dr. Yuetsu Tanaka (University of the Ryukyus) [50]. MT-2 or Jurkat E6.1 cells were prepared to a density of 2.0 × 10^3^ cells/20 µL, mixed with 0, 5, 10, and 25 μg/mL of mAb, and incubated for 1 h at 37 °C. Mouse IgG (25 μg/mL) was used as a negative control. Following incubation, the cells were mixed with Jurkat LTR-sec NL prepared at a concentration of 1.0 × 10^4^ cells/100 µL and incubated at 37 °C for 48 h in 5% CO_2_. After incubation, secNL was determined using a Nano-Glo luciferase assay (Promega).

### 2.13. HTLV-1 gp46 mAb Internalization Assay

MT-2 cells were incubated with 10 μg/mL of the antibody for 1 h at 4 °C. Next, the cells were washed twice with wash buffer [containing 2% (*v*/*v*) FBS and 0.02% (*w*/*v*) NaN_3_ in PBS]. Thereafter, cells with antibodies bound to their surface were incubated in 10% FBS-RPMI medium for 0, 1, 2, 4, or 6 h at 37 °C in 5% CO_2_. Following incubation, the cells were stained with PE goat anti-mouse IgG or PE goat anti-rat IgG for 1 h on ice. The cells were then fixed and subjected to FACS analysis as described before.

### 2.14. Generation of IR700 Conjugated mAb

The antibody against HTLV-1 Env and Human parainfluenza virus 3 (HPIV3) Env were conjugated to IR700 using the IRDye 700DX protein labeling kit (LI-COR, Lincoln, NE, USA). Briefly, 1 mg mAb was incubated with 7 μL IR700 in 0.1 M K_2_HPO_4_ (pH 8.5) at room temperature for 2 h. The mixture was purified using Pierce Zeba Desalting Spin Columns (Thermo Fisher Scientific), and the concentrations of the antibodies and IR700 were determined at 280 nm and 689 nm, respectively. The dye/protein ratio was calculated, and approximately two IR700 molecules were found to be bound to one antibody.

### 2.15. Cell Viability and Lactate Dehydrogenase (LDH) Assay

MT-2, Jurkat and MT-1 cells were incubated with Clone D-IR700 and LAT-27-IR700 (0.01–1 μg/mL) at 37 °C for 1 h and irradiated with 10 J/cm^2^ of NIR light. After incubation for 1 h atRT, Cell titer-Glo (Promega) and LDH-Glo (Promega) were used for the cell viability and LDH assays, respectively. Each assay was performed according to the manufacturer’s protocol. Fluorescence was detected using a Glo-MAX plate reader (Promega). 

### 2.16. Live/Dead Cell Staining Assay

MT-2 cells were incubated with Clone D-IR700 mAb and LAT-27-IR700 (1 μg/mL) at 37 °C for 1 h and irradiated with 10 J/cm^2^ of NIR light. After incubation for 1 h at 37 °C, the MT-2 cells were washed three times with PBS and stained using the Live/Dead Cell Staining kit II (PromoKine, Baden-Württemberg, Germany) for 1 h at RT. Next, the cells were washed three times with PBS and subjected to fluorescence microscopy (BZ-X810, Keyence).

### 2.17. Reporter Assay after NIR-PIAS

MT-2 cells were prepared to a density of 2.0 × 10^3^ cells/20 µL, mixed with 1 μg/mL of clone D-IR700, and incubated for 1 h at 37 °C. Following incubation, the cells were irradiated with 10 J/cm^2^ of NIR light and co-cultured with Jurkat LTR-sec NL prepared at a concentration of 1.0 × 10^4^ cells/100 µL at 37 °C for 48 h in 5% CO_2_. After incubation, secNL was determined using a Nano-Glo luciferase assay (Promega). Human parainfluenza virus 3-IR700 (HPIV3-IR700) was used as negative control.

### 2.18. Statistical Analysis

Statistical analysis was performed using a two-tailed Mann–Whitney U test in GraphPad Prism software version 7.02 (GraphPad Software Inc. San Diego, CA, USA). For all statistical analyses, differences were considered to be statistically significant at *p* < 0.05.

## 3. Results

### 3.1. Synthesis of the HTLV-1 Env Protein and Production of mAbs

To generate mAbs, we synthesized HTLV-1 Env using a wheat germ cell-free protein synthesis system, which produces proteins of precise conformation [44,45]. First, the HTLV-1 Env encoding gene was cloned into the pEU-His vector, which enabled the expression of His-tagged proteins in the wheat germ cell-free system with azolectin liposomes (Figure 1A). The synthesized HTLV-1 Env proteins anchored in the liposomes were separated by Accudenz gradient centrifugation and detected using CBB staining (Figure 1B). The purified proteins were then used to immunize BALB/c mice. Four weeks after immunization, splenocytes were isolated from mice, and four clones (designated A–D) were chosen for further investigation based on their reactivity against HTLV-1 Env (data not shown). The purified mAbs from the four hybridoma clones were tested for reactivity against HTLV-1-infected MT-2 cells by flow cytometry without prior fixation and permeabilization and were found to bind to the extracellular domain (gp46) of the envelope protein on the cell surface (Figure 1C). These mAbs were also found to be applicable in immunostaining (Figure 1D). Isotype analysis revealed that Clone A, B, and C mAbs belonged to the IgG1 kappa isotype and Clone D to the IgG2b kappa isotype (Figure 1E). The mAbs were also tested by ELISA for their reactivity with the HTLV-1 gp46 protein. Clone D showed significantly higher reactivity to the antigen (Figure 1F) and was therefore selected for further functional analysis.

### 3.2. Epitope Analysis of Selected mAbs against HTLV-1 Env

Since mAb Clone D showed the highest reactivity, we set out to determine its binding site on HTLV-1 Env protein gp62, using deletion mutants. The first immunoblot was performed on two mutant variants harboring mutations in the gp46 and gp21 regions, which constitute the HTLV-1 Env (Figure 2A). The HTLV-1 gag protein, p19, was used as a negative control. The results showed that mAb Clone D recognized the gp46 region of HTLV-1 Env, consistent with the FACS analysis. Although gp46 has four glycosylation sites [51], glycosylation is unlikely to occur in the wheat cell-free system [52]. The band shift of gp46 up to 63 kDa might be due to other protein modifications. Next, three gp46 mutants harboring deletions in the receptor-binding domain (RBD), proline-rich domain (PRR), and C-terminal domain (CTD) were prepared. The immunoblotting results showed that the mAb Clone D was bound to the PRR region of gp46 (Figure 2B). Next, we performed an AlphaScreen assay to map the epitope region (Figure 2C). Ten sequential N-terminal biotinylated peptides (PRD1–PRD10) that span the entire length of the PRR were used. We found that mAb Clone D bound to PRD-2 (LPPTAPPLLP) and PRD-3 (TAPPLLPHSN), whose overlapping region was the estimated epitope, spanning amino acid residues 186–192 (Figure 2D). A three-dimensional structure prediction model was generated from the HTLV-1 gp46 epitope to predict the binding site of mAb Clone D. Although it was a predictive model, mAb Clone D was found to recognize the outer part of the gp46 protein, suggesting that the binding site may be easily accessible under biological conditions (Appendix A). To characterize the K_D_ of mAb Clone D and the gp46 protein, we used the BLI Octet Assay System as previously reported [48]. The K_D_ for antibody–antigen binding in Clone D was observed to be very low (<1 × 10^−12^), suggesting a high affinity of this mAb for HTLV-1 gp46 (Figure 2E).

### 3.3. Quantitative HTLV-1-Mediated Cell Fusion Assay Using secNL

We next determined whether mAb Clone D had neutralizing activity to inhibit viral infection since amino acid residues 191–196 of HTLV-1 Env represent the recognition site of LAT-27, a well-known neutralizing antibody [50] (Figure 3A). To date, HTLV-1 infection has generally been assessed based on syncytium formation [20]. However, syncytium formation cannot be quantified. Therefore, we developed a new system for assessing viral transmission by modifying a method described by Alais et al. [49]. Using lentiviral vectors, we generated Jurkat cells stably transfected with LTR-secNL, which can be activated in a Tax-dependent manner upon infection with HTLV-1 (Figure 3B). HTLV-1 infected MT-2 cells were co-cultured with Jurkat LTR-secNL cells in the presence or absence of mAbs, and viral neutralization was measured by quantifying NanoLuc luciferase activity to monitor cell-to-cell infection. We used a neutralizing LAT-27 mAb as a positive control. LAT-27 reduced the signal in a concentration-dependent manner and inhibited HTLV-1 infection, whereas mAb Clone D did not inhibit cell-to-cell infection (Figure 3C), indicating that the mAb Clone D is non-neutralizing.

### 3.4. Internalization Assay of mAb against Anti-HTLV-1 Env 

Neutralizing antibodies are rapidly internalized in HIV-infected cells via binding interactions with “closed” Env trimers [53]. Although the structure and fusion mechanism of HTLV-1 Env remain unclear, we predicted a similar mechanism in HTLV-1-infected cells and investigated the antibody internalization of HTLV-1-infected cells. HTLV-1-infected MT-2 cells were cultured with LAT-27 or mAb Clone D to allow binding to their cell surface. After culturing for 0–6 h, cells were stained with secondary antibodies and the presence of cell surface-bound antibodies was confirmed by flow cytometry. Because the mAb Clone D is non-neutralizing, it was assumed that the antibody is retained longer on the cell surface than LAT-27 (Figure 3D). As expected for LAT-27, the percentage of antibody-positive cells on the cell surface decreased with increasing incubation time, suggesting antibody internalization. This phenomenon was significantly lower in mAb Clone D, which remained on the cell surface for longer times (Figure 3E,F). Furthermore, the internalization rate of mAb into cells was confirmed using confocal microscopy, showing that the internalization rate of Clone D was prominently lower than that of LAT-27 at 37 °C (Figure 3G).

### 3.5. Application of Antibodies to NIR-PIAS

Our results indicated that mAb Clone D bound the outer part of the gp46 protein with high affinity and relatively low internalization. We recently developed an NIR-PIAS targeting pathological microorganisms. This strategy uses pathogen-specific mAbs conjugated to a photoplastic phthalocyanine-derivative IR700 [39]. The photo-activated probe can induce certain structural changes following irradiation with NIR light, leading to mechanical stress that damages membrane structures in proximity to the antibody-binding sites. Using NIR-PIT, it has been reported that the intracellular uptake of IR700 binding antibodies reduces NIR irradiation-induced cytotoxicity [54]. However, because mAb Clone D showed low intracellular uptake, we proceeded to investigate its application in NIR-PIAS. MT-2 cells expressing HTLV-1 Env bind to mAb-conjugated IR700, causing NIR irradiation to damage the cell membrane and resulting in cytotoxicity and leakage of cell contents. Jurkat cells showed no binding of mAb-IR700 and no cytotoxicity after NIR irradiation (Figure 4A). First, we generated Clone D and LAT-27 antibody conjugates with IR700 and checked the conjugation efficiency. We found that both mAb Clone D and LAT-27 were adequately labeled with IR700 (Appendix A). Next, we examined the efficacy of NIR-PIAS by assessing the cytotoxicity exhibited by these antibody conjugates (Clone D-IR700 and LAT-27-IR700) to HTLV-1-infected MT-2 cells using cell viability and LDH assays. Clone D-IR700 or LAT-27-IR700 was added to MT-2 or Jurkat cells and incubated at 37 °C for 1 h, followed by irradiation with 10 J/cm^2^ of NIR light. Cell viability and cytotoxicity were determined 1 h after irradiation. The cell viability assay showed that Clone D-IR700 caused cytotoxicity in a concentration-dependent manner in MT-2 cells (Figure 4B) but not in Jurkat cells (Appendix A). The cytotoxicity of LAT-27-IR700 by NIR irradiation was substantially weaker than that of Clone D-IR700. NIR irradiation also induced cytotoxicity in MT-1 cells, an ATL-derived cell line (Appendix A). NIR-PIT induces necrosis-like cell death, causing LDH release into the cells to which the antibody binds [24]. The LDH assay showed an increase in LDH in an antibody concentration-dependent manner with Clone D-IR700 in MT-2 cells (Figure 4C) but not Jurkat cells (Appendix A). Next, we performed cell staining to examine the effect of NIR-PIAS. The results showed that cell death was induced by NIR irradiation of Clone D-IR700-bound cells (Figure 4D and Appendix A). Finally, we performed reporter assays using HTLV-1 positive cells to examine whether NIR-PIAS inhibits HTLV-1 infection. NIR irradiation of MT-2 cells conjugated with Clone D-IR700 significantly increased LDH in the culture supernatant. When the cells were co-cultured with LTR-secNL Jurkat, secNL signaling was decreased, suggesting that NIR-PIAS suppressed HTLV-1 infection by inducing cytotoxicity (Figure 4E). These results suggest that Clone D-IR700 can be readily applicable to NIR-PIAS for efficient killing of HTLV-1-infected cells.

## 4. Discussion

HTLV-1 infection is widespread, with treatment focusing only on alleviating the symptoms of ATL and related diseases caused by HTLV-1 (e.g., HAM/TSP and HU). Currently, no specific antiviral treatment is available for HTLV-1. Moreover, reports have suggested that the HTLV-1 proviral load is a risk factor for associated diseases [5], and reducing the amount of HTLV-1 may not only prevent the development of these associated diseases but also prevent transmission to others. Here, we hypothesized that the antibody-mediated depletion of HTLV-1-infected cells is an effective therapeutic tool.

The HTLV-1 Env antigen was generated using a wheat cell-free protein synthesis system, which synthesizes precisely folded biologically active proteins similar to those expressed in mammalian cells [55,56]. Moreover, we used a technique known as proteoliposomes, which embeds proteins on the surface of liposomes. Integration of membrane proteins into liposomes supports the proper folding of transmembrane domains and retains their activity and solubility [57]. This method is suitable for the preparation of immunogens to create antibodies that target membrane proteins [58]. 

Recent studies have shown that HTLV-1 infection is mediated by the interactions of Env with three different molecules: neuropilin 1 (NRP1), heparan sulfate proteoglycan (HSPG), and glucose transporter-1 (GLUT-1). At first, gp46 utilizes NRP1 and HSPG to attach and bind to target cells [59,60]. After the binding of gp46 to HSPG/NRP1, gp46 undergoes a conformational change, exposing a binding domain that interacts with GLUT-1 [61]. Then, gp46 binds to GLUT-1, which is accompanied by the isomerization of disulfide linkages between gp46 and gp21, and triggers a conformational change in gp21 [62,63]. The N-terminal hydrophobic fusion peptide of gp21 is inserted into the target cell membrane, and membrane fusion proceeds through three phases of lipid mixing or hemifusion, pore formation, and pore dilation [64]. An antibody against HTLV-1 Env obtained in this study (Clones D) recognized the PRR region, similar to LAT-27 and other neutralizing antibodies [50,65]. This region enables the conformational change of Env during receptor binding, which allows viral entry by promoting the fusion of the viral membrane with the cell membrane by the TM region [12]. These results suggested that the PRR region of HTLV-1 Env is highly antigenic and has profound functionality. Because the obtained antibodies recognized epitopes similar to those of LAT-27, it was expected that these antibodies would also show neutralizing activity; however, they did not.

Since the antibody produced in this study does not have neutralizing activity, we investigated whether it could be used in NIR-PIAS. Clone D-IR700 induced necrosis-like cell death upon NIR irradiation, while LAT-27-IR700 induced only minimal cell death. This could probably be due to differences in the binding mode and affinity of LAT-27 to gp46 compared with those of Clone D. HIV-neutralizing antibodies have been reported to promote the internalization of Env on the cell surface, which is facilitated by neutralizing antibodies that bind to the conformation-dependent “closed” HIV Env trimer [53]. Although the structure of gp46 has not yet been solved, the function of each domain has been predicted based on sequence homology with the Env protein of the gamma retrovirus Murine Leukemia Virus. The PRR region has been suggested to be involved in the conformational change of gp46 following its receptor binding [66]. However, the mechanistic role of this conformational change in Env internalization and membrane fusion during HTLV-1 infection is not well understood. Since the binding sites for Clone D and LAT-27 are very close to each other, the difference in the internalization and/or internalization properties may be attributed to processes following the conformational change of the PRR region in gp46.

The type of cell death induced by NIR-PIT is immunogenic cell death (ICD), which, unlike apoptosis, occurs in a shorter time [67]. Upon ICD, intracellular substances such as LDH and HMGB1 are released from cytoplasm [68]. Because HMGB1 is a major factor for damage-associated molecular patterns (DAMPs), NIR-PIAS might induce a DAMP-induced innate immune response, resulting in the facilitation of antigen-specific immunity targeting pathogenic viruses and virus-infected cells. However, our current study was conducted only in a cell culture model; further in vivo study is necessary to prove the effect of NIR-PIAS in immunity.

Neutralizing antibodies have been developed to prevent HTLV-1 infection [21,22]. Notably, our current NIR-PIAS method can target and eliminate infected cells. We expect a synergistic effect when this approach is combined with neutralizing antibodies. However, NIR-PIAS targets the Env gp46 of HTLV-1, the efficacy of this approach is dependent on the cell surface expression of Env on HTLV-1. Moreover, it is believed that ATL cells exhibit low viral protein expression along with genetic alteration and epigenetic silencing [69], which can potentially attenuate the effect of NIR-PIAS. Recently, single-cell analysis has indicated increased expression of cell surface proteins such as CD73 and CD99 in ATL cells [70]. These proteins may be novel therapeutic targets for ATL cells with defective proviruses devoid of Env expression. In the future, it would be possible to kill ATL cells by targeting these surrogate markers to achieve NIR-PIAS in combination with anti-gp46 mAb.

To clarify the clinical applicability of NIR-PIAS to HTLV-1 therapy, further studies using multiple ATL cell models and humanized mice should be conducted [71]. In particular, cross-reactivity to host proteins with sequences homologous to the binding region of the antibody should be investigated. In fact, the epitope sequence within gp46 has a subtle homology to the human carboxyl terminal portion of osteoprotegerin, which is an osteoclast suppressor protein [72]. The possibility of molecular mimicry should be evaluated to avoid the unnecessary off-target effects of NIR-PIAS. Despite these limitations, NIR-PIAS targeting HTLV-1 Env appears to possess the potential to be developed into a therapeutic option for HTLV-1. Considering the unavailability of effective preventive or therapeutic agents for HTLV-1 infection, further studies should be performed to elucidate the effect of NIR-PIAS in eliminating HTLV-1 infected cells.

## Figures and Tables

**Figure 1 viruses-14-02153-f001:**
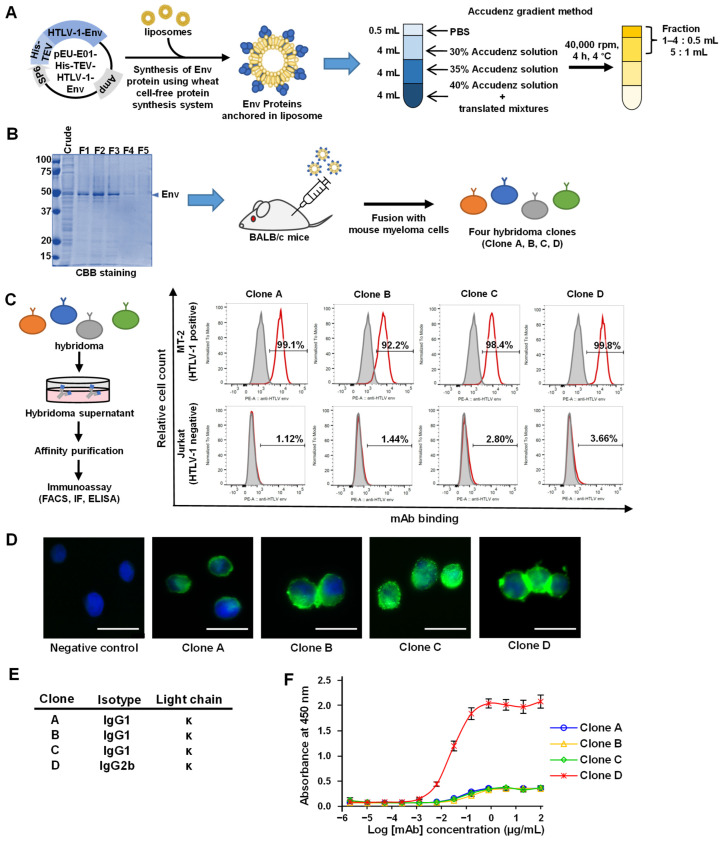
Synthesis of the His-HTLV-1 Env protein and production of anti-HTLV-1 Env mAbs: (**A**,**B**) a scheme showing the production of hybridoma cells to generate anti-HTLV-1 Env mAbs; (**A**) the gene encoding the HTLV-1 Env protein (composed of gp46 and gp21) was cloned into the pEU-His vector and His-HTLV-1 Env proteins anchored in the liposome synthesized using a wheat germ extract cell-free protein synthesis system. Env proteins anchored in liposomes were purified using the Accudenz concentration gradient method; (**B**) purified proteins were visualized by CBB staining, with an arrow indicating the purified proteins in fractions F1–F5. BALB/c mice were immunized by injecting purified HTLV-1 Env proteins. Four weeks after injection, the splenocytes from immunized mice were fused with myeloma cells to establish four hybridoma clones; (**C**) immunoassays performed using the purified mAb from the hybridoma supernatants. Reactivity of the four mAbs to the cell surface of HTLV-1 Env was confirmed by flow cytometry using HTLV-1 positive MT-2 cells. HTLV-1 uninfected Jurkat cells were used as a negative control; (**D**) immunofluorescent analysis of HTLV-1 Env in MT-2 cell. The mAbs of the four clones recognized the cell surface Env protein. Scale bar = 20 μm; (**E**) a list of mAb isotypes produced by the four hybridoma cells; (**F**) dose-dependent binding of purified mAbs to recombinant HTLV-1 gp46 proteins. Data are presented as mean absorbance ± standard deviation (SD) of triplicate assays.

**Figure 2 viruses-14-02153-f002:**
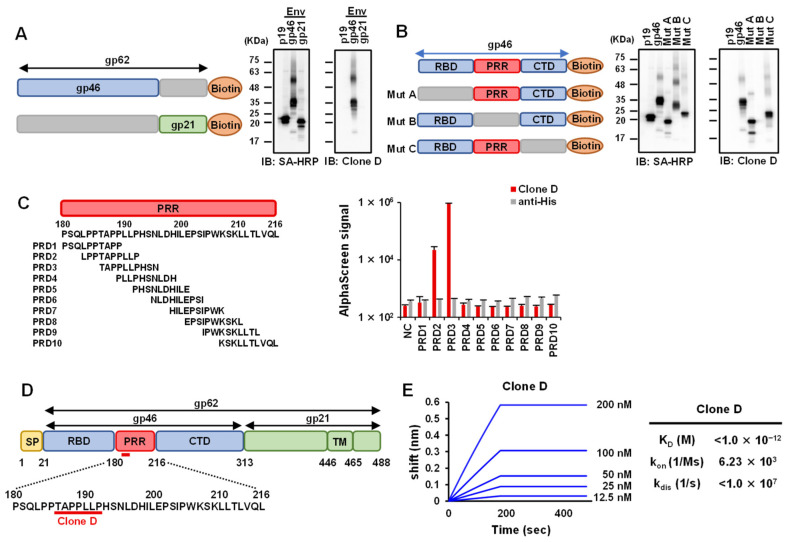
Epitope mapping and affinity measurement of mAb Clone D: (**A**) deletion mutants of HTLV-1 Env gp62 were produced using the wheat germ cell-free protein synthesis system. Reactivity of mAb Clone D to two HTLV-1 Env mutants (gp46 and gp21) was assessed by immunoblotting. Streptavidin (SA)-HRP was used as a positive control. The HTLV-1 gag protein, p19, was used as a negative control; (**B**) epitope mapping of three gp46 mutants, including Mut A (ΔRBD), Mut B (ΔPRR), and Mut C (ΔCTD); (**C**) ten N-termed biotinylated peptides (PRD1–PRD10) were used in the AlphaScreen assay (left). The results of the AlphaScreen assay for mAb Clone D are shown to the right. The anti-His mAb was used as a negative control. NC = Biotinylated dihydrofolate reductase (DHFR) used as a peptide negative control; (**D**) an epitope map of HTLV-1 Env gp62, showing the mAb Clone D recognition site (red line). SP = signal peptide; RBD = receptor-binding domain; PRR = proline-rich region; CTD = C-terminal domain; TM = transmembrane region; (**E**) the K_D_, k_on_, and k_dis_ values of mAb Clone D determined using Bio-Layer interferometry.

**Figure 3 viruses-14-02153-f003:**
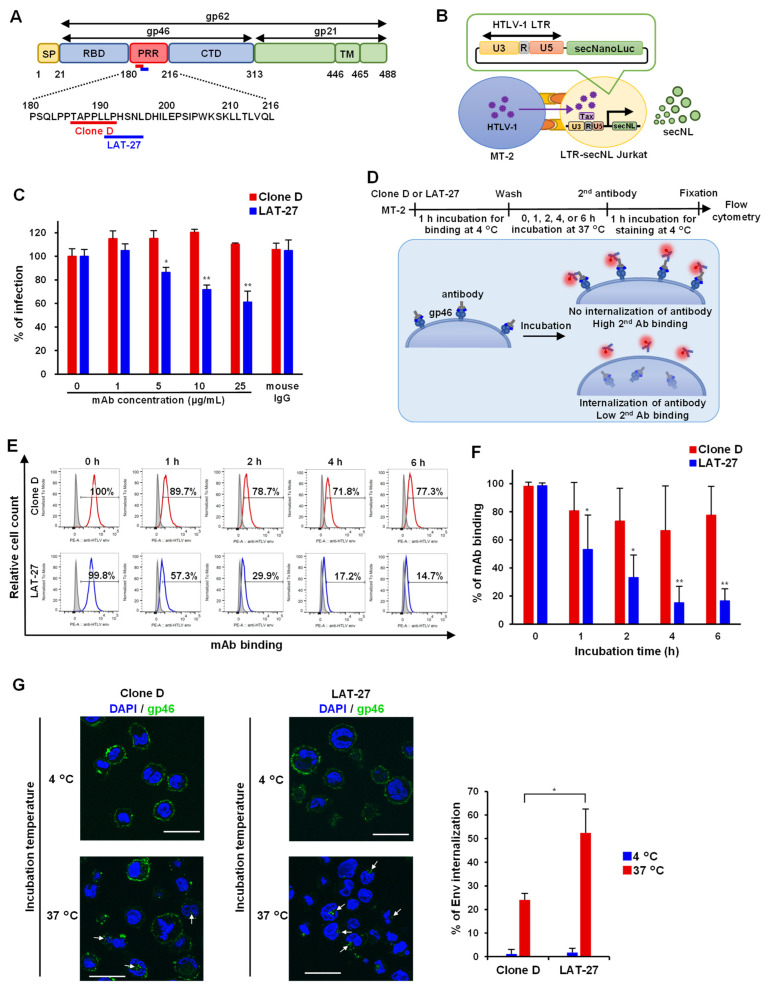
Neutralizing and internalization assays for mAb Clone D and LAT-27: (**A**) an epitope map of HTLV-1 Env gp62, showing the mAb Clone D (red line) and LAT-27 (blue line) recognition sites; (**B**) to establish a reporter assay, LTR-secNL Jurkat cells were generated using a lentivirus. LTR-secNL Jurkat cells were co-cultured with MT-2 cells infected with HTLV-1 and secretion of NanoLuc induced by virus-derived Tax protein; (**C**) the neutralization assay was performed by incubating MT-2 cells with Clone D and LAT-27, mixing them with LTR-secNL Jurkat cells and collecting the supernatant after 48 h. The result shows the percentage of infection when the infection rate at 0 μg/mL of mAb concentration is 100%. Mouse IgG (25 μg/mL) was used as a negative control. Error bars represent the SD of triplicate testing. Significance is indicated by ***p***-values, as follows: * *p* < 0.05, ** *p* < 0.01; (**D**) a scheme explaining the antibody internalization assay, whereby secondary antibody signals (measured by flow cytometry) become reduced following antibody internalization; (**E**) typical results of a triplicate experiment of mAb Clone D and LAT-27 binding of MT-2 cells. The shaded histogram shows non-treated MT-2 cells, and red and blue histograms show Clone D and LAT-27 treated MT-2 cells, respectively; (**F**) antibody internalization rates are displayed as the mean percentage of mAb binding at each time. Error bars represent the SD of triplicate testing. Significance is indicated by *p*-values, as follows: * *p* < 0.05, ** *p* < 0.01; (**G**) confocal microscopy images of MT-2 cells incubated for 1 h at each temperature after binding to HTLV-1 Env mAb. Arrows indicate internalized mAb. Scale bar = 30 μm. The graph shows the percentage of mAb internalization in the confocal microscopy images. Cell internalization rate was calculated by visually counting 8 fields of view. Error bars represent the SD of testing. Significance is indicated by *p*-values, as follows: * *p* < 0.01.

**Figure 4 viruses-14-02153-f004:**
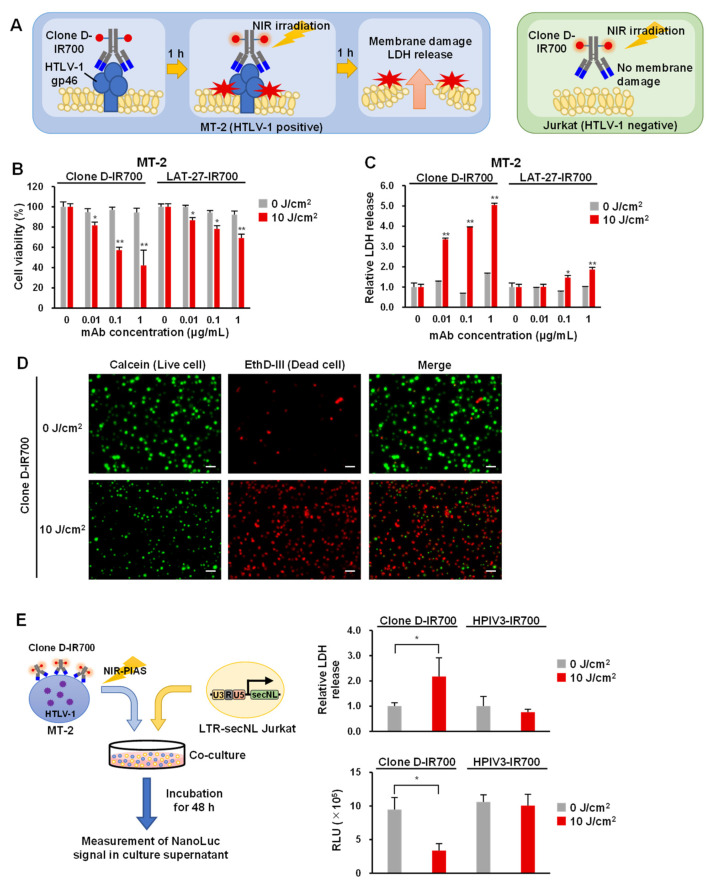
Application of antibodies in NIR-PIAS: (**A**) a schematic of NIR- PIAS, whereby MT-2 cells expressing HTLV-1 Env protein are bound by mAb conjugated IR700, and following NIR light irradiation, cell membranes become damaged, causing cytotoxicity and cell lysis, which releases cell contents (e.g., LDH). Jurkat cells are not bound by antibodies and show no cytotoxicity after irradiation; (**B**) results of the cell viability assays using Clone D-IR700 and LAT-27-IR700 for MT-2 cells. The result shows the percentage of cell viability when the 0 μg/mL of mAb concentration is 100%. Error bars represent the SD of triplicate testing. Significance is indicated by *p*-values, as follows: * *p* < 0.05, ** *p* < 0.01; (**C**) results of the LDH assays using Clone D-IR700 and LAT-27-IR700 for MT-2 cells. The result shows relative LDH release normalized at 0 μg/mL of mAb concentration. Error bars represent the SD of triplicate testing. Significance is indicated by *p*-values, as follows: * *p* < 0.05, ** *p* < 0.01; (**D**) cell staining of MT-2 cells incubated with Clone D-IR700 and LAT-27-IR700 after irradiation with (10 J/cm^2^) or without (0 J/cm^2^) NIR light. Live cells were stained with Calcein (green) and dead cells with EthD-III (red). Scale bar = 50 μm; (**E**) a schematic of reporter assay after NIR-PIAS. MT-2 cells bound by Clone D-IR700 were irradiated with NIR light, and co-cultured with LTR- secNL Jurkat cells (left). Results of the LDH assay and reporter assay after NIR-PIAS (right). The result shows relative LDH release normalized under non-irradiation (0 J/cm^2^) with NIR light. NL signal was measured after 48 h of incubation. HPIV3-IR700 was used as a negative control. Error bars represent the SD of triplicate testing. Significance is indicated by *p*-values, as follows: * *p* < 0.01.

## Data Availability

Not applicable.

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
