# Peer review of "Development of a Monoclonal Antibody Targeting HTLV-1 Envelope gp46 Glycoprotein and Its Application to Near-Infrared Photoimmuno-Antimicrobial Strategy"

_viruses, 2022, doi:10.3390/v14102153_

Round 1

Reviewer 1 Report

In this paper, the authors discovered a HTLV-1 gp46-targeting non-neutralizing monoclonal antibody, and applied it to a near-infrared photoimmuno-antimicrobial strategy that can specifically kill HTLV-1 infected cells in vitro. The findings from this paper can be potentially used to develop therapeutics against HTLV-1. Overall, the manuscript is well-written with minor issues that need to be addressed:

 1. It would be better to provide more background information about HTLV-1 Env protein and NIR-PIT in the introduction. What is the replication cycle of HTLV-1? What are the mortality and morbidity rates of HTLV-1 infections? How HTLV-1 Env mediates membrane fusion? How near-infrared radiation activates the photosensitive dye to induce cell death?

2. The Env protein of HTLV-1 is likely to be a trimer of gp62, however, the authors only predicted the structure of a monomeric gp46 by AlphaFold2, which would be very likely inaccurate because the interactions between gp46 with the other parts of Env protein is missing. It would be better to predict the structure using full-length HTLV-1 Env, or try to find a similar structure in Protein Data Bank that shares high amino acid sequence identity with HTLV-1 Env.

3. HIV neutralizing antibodies were reported to increase the internalization of Env on cell surface, but this is conformation-dependent, specifically, the neutralizing antibodies that bind to “closed” HIV Env trimers are more likely to be internalized. The authors tried to adopt this theory on HTLV-1 Env protein, but I feel it’s not convincing because the structure and the fusion mechanism of HTLV-1 Env are still unclear, there might be different mechanisms for HTLV-1 Env-mediated antibody internalization. To verify if this theory applies to HTLV-1 Env, more evidence will be needed. Please provide a reasonable discussion about this problem.    

Line 192: the software should be “UCSF Chimera”, not “USFC Chimera”. Please properly cite UCSF Chimera, information can be found here: https://www.cgl.ucsf.edu/chimera/1.5.3/docs/credits.html

Line 445: Please double-check the citation here, it seems to be [36].

Author Response

Responses to the comments of Reviewer #1

In this paper, the authors discovered a HTLV-1 gp46-targeting non-neutralizing monoclonal antibody, and applied it to a near-infrared photoimmuno-antimicrobial strategy that can specifically kill HTLV-1 infected cells in vitro. The findings from this paper can be potentially used to develop therapeutics against HTLV-1. Overall, the manuscript is well-written with minor issues that need to be addressed:

Response: We appreciate the reviewer’s valuable and constructive comments. Please find our point-by-point responses below.

  1. It would be better to provide more background information about HTLV-1 Env protein and NIR-PIT in the introduction. What is the replication cycle of HTLV-1? What are the mortality and morbidity rates of HTLV-1 infections? How HTLV-1 Env mediates membrane fusion? How near-infrared radiation activates the photosensitive dye to induce cell death?

Response: We thank the reviewer for this insightful suggestion. Accordingly, we have added information on the morbidity and mortality of patients infected with HTLV-1 as well as on the virus replication cycle and the mechanism of infection in our revised manuscript (lines 48–53, 56–64, and 483–492). We have also included a brief summary of the mechanism by which cellular cytotoxicity is induced following the denaturation of IR700-binding antibodies by near-infrared irradiation (lines 84–90)

  1. The Env protein of HTLV-1 is likely to be a trimer of gp62, however, the authors only predicted the structure of a monomeric gp46 by AlphaFold2, which would be very likely inaccurate because the interactions between gp46 with the other parts of Env protein is missing. It would be better to predict the structure using full-length HTLV-1 Env, or try to find a similar structure in Protein Data Bank that shares high amino acid sequence identity with HTLV-1 Env.

Response: We appreciate the reviewer’s comments. In fact, we attempted to establish a predicted structure model for the gp62 trimer, but this model did not clearly demonstrate the epitope. Therefore, we alternatively constructed the structure of gp46 monomer and applied the obtained model. Unfortunately, the structure of gp46 has not been solved, nor has a protein with a similar structure been identified. Since the structure mapping of the antibody-binding site is important to determine the suitability of mAb for NIR-PIAS, we have depicted this model in supplementary figure S1. We will re-analyze the feasibility of our model when the real structure of gp46 is clarified by cryo-electron microscopy or crystal structure analysis.

  1. HIV neutralizing antibodies were reported to increase the internalization of Env on cell surface, but this is conformation-dependent, specifically, the neutralizing antibodies that bind to “closed” HIV Env trimers are more likely to be internalized. The authors tried to adopt this theory on HTLV-1 Env protein, but I feel it’s not convincing because the structure and the fusion mechanism of HTLV-1 Env are still unclear, there might be different mechanisms for HTLV-1 Env-mediated antibody internalization. To verify if this theory applies to HTLV-1 Env, more evidence will be needed. Please provide a reasonable discussion about this problem.

Response: We appreciate the reviewer's comment on this point. Although the structure of gp46 has not yet been elucidated, the function of each domain has been predicted based on sequence homology with the Env protein of the gamma retrovirus, Murine Leukemia Virus. The proline-rich (PRR) region has been suggested to be involved in the conformational change of gp46 following receptor binding (Lavillette et al. 1998). However, as the reviewer has indicated, the mechanistic role of this conformational change in Env internalization and membrane fusion during HTLV-1 infection is not well understood. Since the binding sites for Clone D and LAT27 are very close, the difference in internalization may be attributed to processes following the conformational change in the PRR region. Further study is needed to resolve this important issue. We have included these issues in the Discussion section of our revised manuscript (lines 504–515).

Line 192: the software should be “UCSF Chimera”, not “USFC Chimera”. Please properly cite UCSF Chimera, information can be found here: https://www.cgl.ucsf.edu/chimera/1.5.3/docs/credits.html

Response: We apologize for the spelling error. UCSF Chimera citation information has been added to the description (lines 213–214).

Line 445: Please double-check the citation here, it seems to be [36].

Response: We apologize for this oversight. The references have been reviewed and corrected.

Reviewer 2 Report

Hatayama et al. developed a mouse monoclonal antibody against HTLV-1 Env and applied it to near-infrared photoimmuno-antimicrobial strategy (PIAS). They found that this new mAb was hardly taken up intracellularly and remained on the cell surface of HTLV-1-infected cells; the antibody conjugated with IR700 was able to recognize and kill HTLV-1 infected MT-2 cells after near-infrared irradiation.

This is an important study that offers a new option for HTLV-1 infection, which is difficult to control. The development of the antibody has been conducted by high quality methods and the molecular evaluation of the antibody appears to be correct. However, the evaluation of the antibody against infected cells was performed on only one infected cell line, so it seems difficult to conclude from this study alone whether the antibody's effectiveness as an antibody can be generalized. 

- The authors have only tested the binding ability, specificity, and cell death induction of the antibody in MT-2. I think that confirmatory experiments using other cell lines and infection models are needed to generalize the effectiveness of this antibody and methodology.

- Will this new methodology work for tumor cells with low viral expression? Please provide some additional insights.

- What about the specificity of the epitope identified in this case? I would like a comparison with the host sequence to be shown.

- I think it is an excellent antibody and methodology. However, what clinical applications could it possibly have? As has already been shown, HTLV-1 has a low efficiency of infection. There is some wonder what clinical benefit could be gained by using it against the HTLV-1 carriers. What it means to reduce new infections should at least be considered within this study.

- How stable is Env expression in infected cells? Most HTLV-1 infected cells express few viral genes other than HBZ. The clonally expanded malignant clones may be deficient in provirus genome. It should be clearly stated what type of infected cells are sensitive against this antibody. The actual benefit gained by this PIAS antibody should also be clarified in the text.

- Please describe how to obtain LAT-27.

Author Response

Responses to the comments of Reviewer #2

Hatayama et al. developed a mouse monoclonal antibody against HTLV-1 Env and applied it to near-infrared photoimmuno-antimicrobial strategy (PIAS). They found that this new mAb was hardly taken up intracellularly and remained on the cell surface of HTLV-1-infected cells; the antibody conjugated with IR700 was able to recognize and kill HTLV-1 infected MT-2 cells after near-infrared irradiation. This is an important study that offers a new option for HTLV-1 infection, which is difficult to control. The development of the antibody has been conducted by high quality methods and the molecular evaluation of the antibody appears to be correct. However, the evaluation of the antibody against infected cells was performed on only one infected cell line, so it seems difficult to conclude from this study alone whether the antibody’s effectiveness as an antibody can be generalized.

Response: We are grateful for the insightful and constructive suggestions by the reviewer. We have revised our manuscript to address these concerns accordingly.

 - The authors have only tested the binding ability, specificity, and cell death induction of the antibody in MT-2. I think that confirmatory experiments using other cell lines and infection models are needed to generalize the effectiveness of this antibody and methodology.

Response: We fully appreciate the careful analysis done by the reviewer. In addition to MT-2 cells, we used MT-1 cells established from the peripheral blood of ATL patients (Miyoshi 1980). We initially examined the expression of Env on MT-1 cells, which was comparable to that of MT-2. Subsequently, we performed NIR-PIAS on MT-1 cells and found similar levels of cytotoxicity as seen in MT-2 (Figure S4). Because NIR-PIAS targets HTLV-1 Env gp46, the expression of Env protein on the surface of HTLV-1-infected cells may affect the efficacy of this strategy. Therefore, further studies using cell lines with various Env expressions are necessary to confirm this supposition.

- Will this new methodology work for tumor cells with low viral expression? Please provide some additional insights.

Response: We thank the reviewer for this comment. Our current study revealed that NIR-PIAS induced cytotoxicity depending on the cell-surface expression of gp46. Therefore, this strategy may not be effective in cells with low Env expression. We have included additional insights in our response on the stability of Env expression in infected cells presented below.

- What about the specificity of the epitope identified in this case? I would like a comparison with the host sequence to be shown.

Response: We thank the reviewer for this comment. We investigated if the epitope sequence (TAPPLLP) of gp46 has any homology to human proteins by tBLASTn analysis, but no significant sequence homology was found. In fact, our immunostaining with Clone D mAb showed extremely low background signal in HTLV-1-negative Jurkat cells. A previous study indicates that osteoprotegerin has a little homology to gp46 and has been reported to have biological effects on the production of antibodies against HTLV-1 (Sagara et al. 2019). However, this is not the epitope region for Clone D. Therefore, it is unlikely that off-target effects will occur when Clone D mAb is used in NIR-PIAS (lines 537–541).

- I think it is an excellent antibody and methodology. However, what clinical applications could it possibly have? As has already been shown, HTLV-1 has a low efficiency of infection. There is some wonder what clinical benefit could be gained by using it against the HTLV-1 carriers. What it means to reduce new infections should at least be considered within this study.

Response: Thank you for your insightful suggestion. Neutralizing antibodies against HTLV-1 can inhibit cell-to-cell infection but cannot eliminate infected cells. NIR-PIAS can induce cytotoxicity in Env-expressing HTLV-1-infected cells and cause them to eradicate themselves. Therefore, we believe that the co-treatment of neutralizing antibodies and NIR-PIAS can suppress viral spreading and eliminate infected cells in HTLV-1-infected individuals. Further careful analysis using an in vivo model is necessary. These issues are described in our revised text (lines 524–528).

- How stable is Env expression in infected cells? Most HTLV-1 infected cells express few viral genes other than HBZ. The clonally expanded malignant clones may be deficient in provirus genome. It should be clearly stated what type of infected cells are sensitive against this antibody. The actual benefit gained by this PIAS antibody should also be clarified in the text.

Response: These are valid concerns. It is believed that ATL cells exhibit low viral protein expression along with genetic alteration and epigenetic silencing (Gaudray et al. 2002), which can potentially attenuate the effect of NIR-PIAS. Recently, single-cell analysis has indicated increased expression of cell surface proteins such as CD73 and CD99 in ATL cells (Koya et al. 2021). These proteins may be novel therapeutic targets for ATL cells with defective proviruses devoid of Env expression. In the future, it may be possible to kill ATL cells by targeting these surrogate markers to achieve NIR-PIAS in combination with anti-gp46 mAb (lines 528–535). In addition, to clarify the applicability of NIR-PIAS for HTLV-1 therapy, future studies using multiple ATL cell models and humanized mice (Tezuka et al. 2014) should be conducted to examine the off-target effects of antibodies and near-infrared light transparency (lines 536–537).

- Please describe how to obtain LAT-27.

Response: We thank the reviewer for this comment. LAT-27 is a gift provided by Dr. Yuetsu Tanaka (University of the Ryukyus). Information on how LAT-27 was obtained has been provided in the Materials and Methods and the Acknowledgements sections (lines 235–236 and lines 564–565, respectively).

Reviewer 3 Report

Comments to the Authors
The manuscript by Hatayama et al. reports on a novel antibody therapeutic strategy with a monoclonal antibody recognizing the PRR in the HTLV-1 Env gp46.
This monoclonal antibody stayed on the surface of HTLV-1-infected cells, instead of no neutralizing activity for HTLV-1 infection.
The authors observed cytotoxicity with LDH release only for HTLV-1-infected cells, suggesting necrosis occurred by near-infrared photoimmuno-antimicrobial strategy
via the binding of the monoclonal antibody.
From a viewpoint of no reliable treatment for HTLV-1-associated diseases, novel strategy for the prevention of HTLV-1 infection would be good news for improvement of public health,
especially in the endemic areas.
However, the authors do not develop this critical aspect of the issue.
1. The journal ‘Viruses’ does not have strict formatting requirements, however, the authors should strive accurate description.
All abbreviations used should be defined at the first appearance.
2. Abstract.
The last sentence of the abstract is unclear. The author mentioned NIR-PIAS could be a new tool for HTLV-1 infection.
This data was not presented in the article result section where it should be.
The data in Figure 3 only showed the monoclonal antibody derived from Clone D stay long on the surface of HTLV-1-infected cells without neutralizing activity.
In Figure 4, near-infrared light irradiation evokes cell death via the monoclonal antibody.
The authors should demonstrate that this antibody as well as NIR is involved in the suppression of HTLV-1 infection.
In this study, only MT-2 cells were subjected, however, MT-2 is the very unique cell line with constant high expression of HTLV-1 proteins on the surface of all cells.
Many of the HTLV-1-infected cells transiently or sparsely express viral proteins on cell surface.
In addition, the ectodomain of HTLV-1 Env proteins depends on the infected cells (cell lines and primary cells from infected individuals).
3. Materials and methods.
Accurate description should be given to ensure reproducibility.
For example,
P.3 L.100 What is ‘epitome’?
P.4 L.192 UCSF Chimera is a program developed by Prof. Thomas Ferrin in UCSF.
P.5 L.231 env means genes coding Env proteins. 4. Results.
The synthetic gene coded full-length of HTLV-1 Env proteins was cloned, and proteins-anchored liposomes synthesized by using wheat germ cell-free system.
In Figure 1B, fraction1-3 showed only one band in each lane at 35 kDa, indicating that the obtained major product might be the HTLV-1 gp46 protein.
On the contrary, in Figure 2A, the product derived from deletion mutant of Env showed 63 kDa band in addition to 35 kDa band in lane of ‘gp46’.
The reason should be mentioned. According to the legends, Figure 3F showed the mean percentage of triplicate exams. If so, what does Figure 3E represent?
Are the patterns and percentages in figures typical ones?
5. Discussion.
Although internalization of LAT27 after binding to HTLV-1 Env on the cell surface are mentioned, from the data in Figure 3, only the disappearance of the antibody molecules was shown.
In order to prove internalization of LAT27, it is necessary to observe the confocal microscopy analysis and the effect of incubation temperature as in Ref. 36.
The authors demonstrated that the cell death induced by NIR-PIT is necrotic and DAMPs are released from the cell to the plasma membrane, however, it was shown HMGB1 had been released into supernatant of necrotic cells in Ref. 46.
6. References.
References must be numbered in order of appearance in the text.
References on NIR-PIT are too biased towards reports from a specific group.
References that are not cited are listed.
P.12 L.445 The authors cited as [35], but the correct citation seems to be [36].

7. Figures.

SI Units (International System of Units) should be described with accurate manner. There is no description about NC of the horizontal axis of the graph in Figure 2C. In Figures 3 and 4, p values were mentioned without any description of statistical analyses in Materials and Methods section.

8. I think that this manuscript should be properly proofread by native English speakers.

Author Response

Responses to the comments of Reviewer #3

Comments and Suggestions for Authors

Comments to the Authors
The manuscript by Hatayama et al. reports on a novel antibody therapeutic strategy with a monoclonal antibody recognizing the PRR in the HTLV-1 Env gp46. This monoclonal antibody stayed on the surface of HTLV-1-infected cells, instead of no neutralizing activity for HTLV-1 infection. The authors observed cytotoxicity with LDH release only for HTLV-1-infected cells, suggesting necrosis occurred by near-infrared photoimmuno-antimicrobial strategy via the binding of the monoclonal antibody. From a viewpoint of no reliable treatment for HTLV-1-associated diseases, novel strategy for the prevention of HTLV-1 infection would be good news for improvement of public health, especially in the endemic areas. However, the authors do not develop this critical aspect of the issue.

Response: We thank the reviewer for their insightful and constructive remarks. We have revised the manuscript to address these concerns based on their valuable suggestions.

  1. The journal ‘Viruses’ does not have strict formatting requirements, however, the authors should strive accurate description. All abbreviations used should be defined at the first appearance.

Response: We have double-checked and corrected the formatting of all abbreviations.

  1. Abstract.
    The last sentence of the abstract is unclear. The author mentioned NIR-PIAS could be a new tool for HTLV-1 infection. This data was not presented in the article result section where it should be. The data in Figure 3 only showed the monoclonal antibody derived from Clone D stay long on the surface of HTLV-1-infected cells without neutralizing activity. In Figure 4, near-infrared light irradiation evokes cell death via the monoclonal antibody. The authors should demonstrate that this antibody as well as NIR is involved in the suppression of HTLV-1 infection.

Response: In accordance with the reviewer's suggestion, we examined whether NIR-PIAS could suppress HTLV-1 infection. MT-2 cells were subjected to NIR-PIAS and then co-cultured with NanoLuc Jurkat cells to measure cell-to-cell infection. (Figure 4E left). We found that the NanoLuc signal was prominently reduced in cells treated with NIR-PIAS, indicating that NIR-PIAS can suppress HTLV-1 infection (Figure 4E right).

In this study, only MT-2 cells were subjected, however, MT-2 is the very unique cell line with constant high expression of HTLV-1 proteins on the surface of all cells. Many of the HTLV-1-infected cells transiently or sparsely express viral proteins on cell surface. In addition, the ectodomain of HTLV-1 Env proteins depends on the infected cells (cell lines and primary cells from infected individuals). 

Response: We thank the reviewer for this comment. In addition to MT-2 cells, MT-1 cells were used in our study. Please see the responses to the comments of Reviewer #2.

  1. Materials and methods.
    Accurate description should be given to ensure reproducibility. For example, P.3 L.100 What is ‘epitome’?

Response:We agree with the reviewer's comment and have accordingly changed the word “epitome” to “epitope.”

P.4 L.192 UCSF Chimera is a program developed by Prof. Thomas Ferrin in UCSF.

Response: We thank the reviewer for this comment. Information on the UCSF Chimera citation has been added to the text (lines 213–214).

P.5 L.231 env means genes coding Env proteins. 

Response: We apologize for this oversight and have accordingly corrected it.

  1. Results.
    The synthetic gene coded full-length of HTLV-1 Env proteins was cloned, and proteins-anchored liposomes synthesized by using wheat germ cell-free system. In Figure 1B, fraction1-3 showed only one band in each lane at 35 kDa, indicating that the obtained major product might be the HTLV-1 gp46 protein. On the contrary, in Figure 2A, the product derived from deletion mutant of Env showed 63 kDa band in addition to 35 kDa band in lane of ‘gp46’. The reason should be mentioned.

Response: We appreciate the careful analysis by the reviewer. We checked the molecular weight marker for CBB staining and found that it was incorrectly described. We have amended this error in our revised Figure 1B. After correcting the molecular weight marker, the band size of cell-free synthesized recombinant gp62 was observed around 50 kDa. Although gp46 has four glycosylation sites (Seiki et al. 1983), glycosylation is unlikely to occur in the wheat cell-free system (Harbers 2014). The band shift of gp46 up to 63 kDa might be due to other protein modifications. We have described this issue in our revised manuscript (lines 313–315).

According to the legends, Figure 3F showed the mean percentage of triplicate exams. If so, what does Figure 3E represent? Are the patterns and percentages in figures typical ones? 

Response: We appreciate the reviewer’s comment on this point. Figure 3E presents a representative histogram of triplicate examinations. We have clearly described this issue in the legend of Figure 3E.

  1. Discussion.
    Although internalization of LAT27 after binding to HTLV-1 Env on the cell surface are mentioned, from the data in Figure 3, only the disappearance of the antibody molecules was shown. In order to prove internalization of LAT27, it is necessary to observe the confocal microscopy analysis and the effect of incubation temperature as in Ref. 36.

Response: We wish to thank the reviewer for this comment. According to the reviewer’s suggestion, an additional analysis with confocal microscopy was performed to confirm the antibody internalization. We found the punctate structure in the cytoplasm resulting from antibody internalization at 37°C (Figure 3G left). After treating cells with either Clone D or LAT-27 antibody, we found that the internalization rates were changed along with incubation temperature (Figure 3G right). Clone D mAb showed a lower rate of antibody internalization than LAT-27 (Figure 3G right). The rate of antibody internalization into the cells was calculated by visually counting 8 fields of view. We described these issues in the figure legend (lines 411–416).

The authors demonstrated that the cell death Induced by NIR-PIT is necrotic and DAMPs are released from the cell to the plasma membrane, however, it was shown HMGB1 had been released into supernatant of necrotic cells in Ref. 46. 

Response: We thank the reviewer for this comment. NIR-PIAS is a method of inducing cell injury together with the aggregation of antibody-bound membrane proteins by the near-infrared light irradiation (Kobayashi et al. 2021). The increase in the cell supernatant LDH is caused by the leakage of cytoplasmic LDH following the damage of the plasma membrane. Literature indicates that HMGB1 is also released upon necroptotic cell death (Xia et al. 2021). Although we did not measure HMGB1, it is likely that DAMPs including HMGB1would be increased in the cell supernatant following NIR-PIAS. To avoid further confusion on part of the readers, we have changed the reference for the release of LDH and HMGB1 (References 68).

  1. References.
    References must be numbered in order of appearance in the text. References on NIR-PIT are too biased towards reports from a specific group. References that are not cited are listed.P.12 L.445 The authors cited as [35], but the correct citation seems to be [36].

Response: We thank the reviewer for this comment. The citations have been corrected to be in the order listed. We have added references on NIR-PIT reported by different groups (References 28, 30 and 36) and checked that the text corresponds to the references cited.

  1. Figures.

SI Units (International System of Units) should be described with accurate manner. There is no description about NC of the horizontal axis of the graph in Figure 2C. In Figures 3 and 4, p values were mentioned without any description of statistical analyses in Materials and Methods section.

Response: We thank the reviewer for this comment. The units in the text and figures were reconfirmed and corrected. We apologize for not describing the negative control in detail. Indeed, biotinylated DHFR was used as a peptide negative control in Figure 2C. We have described this issue in both the Materials and Methods and the Figure Legends sections. To avoid confusion on part of the readers, we also changed the negative control for the antibody from NC to anti-His (Figure 2C). Statistical analysis has been added to the Materials and Methods section to provide details on the data analysis methods (lines 283–286).

  1. I think that this manuscript should be properly proofread by native English speakers.

Response: The manuscript has been reviewed by native English speakers through a professional English language-editing service.

Round 2

Reviewer 2 Report

All my concerns have been addressed in the revised manuscript. I think this is an important study that should be published in this journal.

Reviewer 3 Report

When indicating the number of lines in your revised version in the responses to the comments of reviewers, you should finally check the uploaded version.